# Perceived stress and associated factors among primary caregivers of patients with mental illness attending the Outpatient Department at Jimma Medical Center, Southwest Ethiopia: A cross-sectional study

**Nuguse Daraje***, **Susan Anand**, **Tilahun Legese**

Faculty of Health sciences, School of Nursing, Jimma University, Jimma, Oromia, Ethiopia

☉ These authors contributed equally to this work.
* nugusedaraje@gmail.com

## Abstract

Mental illness is a major global burden affecting millions of people and straining health, economies, and families. Primary caregivers of patients with mental illness often experience significant problems with emotional, physical, and financial issues due to the demands of their role, resulting in perceived stress, which affects their well-being and the quality of care provided. Despite this, there is inadequate information on their perception of caregiver stress in Ethiopia. This study aimed to assess perceived stress and associated factors among primary caregivers of patients with mental illness attending Jimma Medical Center. A cross-sectional study was conducted on 409 primary caregivers at the Jimma Medical Center psychiatric outpatient department from May 6 to June 7, 2024. The study sample was selected using the convenience sampling method. Data were collected using the Perceived Stress Scale-10 (PSS-10) questionnaire through face-to-face interviews and analyzed using SPSS version 26. Bivariable and multivariable ordinal logistic regression analyses were done to identify factors associated with perceived stress at P-value ≤ 0.05. Among 409 primary caregivers, the prevalence of perceived stress was 13.2% high, 62.1% moderate, and 24.7% low. Lack of formal education among caregivers (OR=2.236, CI: 1.132-4.419), patient aggressive behavior (OR=2.315, CI: 1.310-4.092), high care burden (OR=4.011, CI: 2.083-12.938), low coping strategies (OR=3.611, CI: 1.223-9.366), and patient comorbid illnesses (OR=3.074, CI: 2.481-10.077) were statistically associated with caregivers' perceived stress. Three-fourths of primary caregivers of patients with mental illness experience moderate to high stress, while one-fourth perceive low stress. The high prevalence of stress, particularly among less-educated caregivers; high care burden; low coping strategies; and those managing patient aggression, improved access to mental health resources,

**Data availability statement :** All relevant data are within the paper and its Supporting Information files.

**Funding:** The author(s) received no specific funding for this work.

**Competing interests:** The authors have declared that no competing interests exist.

**Abbreviations:** CI: Confidence interval, IRB: Institutional Review Board, JMC: Jimma Medical center, JU: Jimma University, OR: Odds Ratio, OPD: Outpatient Department, OSSS: Oslo Social Support Scale, PSS: Perceived Stress Scale, PTSD: Post-traumatic stress disorder, SPSS: Statistical Package for Social Science, VIF: Variance Inflation Factor, WHO: World Health Organization, ZBI: Zarit Burden Interview.

building resilience programs, coping strategies, and enhancing social support networks are needed to mitigate caregiver stress and improve overall well-being.

## Introduction

Mental illness is a health condition that involves changes in emotion, thinking, or a combination of these and is associated with distress or difficulties functioning in social, work, or family activities [1]. Its global burden affects 970 million people worldwide [2]. This condition poses significant challenges for both patients and their primary caregivers, who often bear the emotional, physical, and financial strains from providing daily support [3,4] . Perceived stress is the subjective feeling of being overwhelmed or unable to cope with the demands of a situation, which differs from person to person based on their perceptions of a situation, such as their views about their ability to cope or the potential consequences of the event [5]. Family members provide the majority of the care in many cultures, which creates a special set of stressors that can affect their wellbeing [6–8].

Primary caregivers play a crucial role in supporting patients with mental illness conditions because they deliver most informal care to patients. but often face high perceived stress due to the nature of the illness, societal stigma, financial strain, and inadequate support systems [9–11]. Some studies show that primary caregivers for patients with mental illness frequently deal with high levels of perceived stress and burden of care; the severity of the patient's illness; the type of mental illness; the length of time spent providing care; a lack of social support, financial strain, social isolation and stigma; emotional impact; coping mechanisms; and resilience [12–21].

Therefore, understanding the perceived stress of primary caregivers and associated factors is crucially significant to plan family intervention programs that can alleviate their stress and enhance patient care outcomes [22]. In Ethiopia, where mental health services are limited and families provide the majority of support, target interventions to reduce stress levels and improve overall primary caregiver well-being and improve patient outcomes. Therefore, this study is aimed to fill this gap by assessing perceived stress and associated factors among primary caregivers of patients with mental illness attending the psychiatric outpatient department at Jimma Medical Center.

## Method and materials

### Ethics statement

Ethical approval for the study was obtained from the Institutional Review Board (IRB) of Jimma University with Ref. No.: JUIH/IRB/157/24. After ethical clearance was received to get permission to conduct the research, an official letter was written for support from nursing school to Jimma Medical Center. The involvement of the study participants was voluntary, and participants were informed of the right to withdraw anytime from the study. The objective of the study was clearly communicated with study participants, and data was collected after written informed consent

was obtained from all participants prior to data collection. To ensure privacy and confidentiality, all information and data obtained from study participants were kept confidential.

## Study design and setting

A cross-sectional study was conducted from May 6 to June 7, 2024, at Jimma Medical Center, which is in the Jimma Zone, Oromia region. It is located 352km southwest of the capital city of Ethiopia, Addis Ababa. Jimma Medical Center currently provides service for more than 15 million people living in southwest Ethiopia. The psychiatry clinic at Jimma Medical Center was established in 1988.

## Study participants

The minimum sample size required for this study was calculated by using the single population proportion formula. $n = \frac{(Z\alpha/2)^2 p(1-p)}{d^2}$  $n = \frac{(1.96)^2 0.5(1-0.5)}{(0.05)^2}$, $n = 384$. Then, by adding 10% of the non-respondent rate, which is 38, the total sample size for this study was $384 + 38 = 422$. Out of the total sample size of 422 primary caregivers of mental illness who have been followed up at a psychiatric outpatient department for at least 6 months, 409 had provided care. Participants were selected using a convenience sampling technique due to feasibility constraints during psychiatric follow-up treatment. Caregivers who fulfilled the inclusion criteria were included in the study sequentially until the final sample size was reached. In the event that the patient was accompanied by more than one person, the person who did the major share of caregiving was selected.

## Data collection

Data collection tools were prepared after reviewing relevant literature. Conducted face-to-face interviews using validated tools, and the instrument consisted of nine parts, including the Perceived Stress Scale (PSS-10) [23], Zarit Burden Interview (ZBI) [24,25], the Oslo Social Support Scale [26–28], Modified Consumer Experiences of Stigma Questionnaires [29,30], the Coping Scale [31], the Brief Resilience Scale (BRS) [32], and the Alcohol, Smoking and Substance Involvement Screening Test (ASSIST) [33] . Socio-demographics and clinical characteristics were collected from caregivers and patient records.

The questionnaire was prepared first in English and translated into the local language with back translation to English to check the consistency. Standardized and validated tools were used in this study. To identify potential problems and to make important modifications, the questionnaire was pretested on 5% of the sample size (n = 19) one week before actual data collection at Shanan Gibe General Hospital, and it gives reliability (Cronbach's alpha was 0.78) for the Perceived Stress Scale-10 (PSS-10). Based on the pre-test result, appropriate corrections were made, such as logical order and socio-demographic characteristics questions, and contextualization was made according to primary caregiver context. The collected data were checked for completeness and consistency by the principal investigator and supervisor every day at the end of each data collection day, and if necessary, corrective measures were made for the area where difficulties were identified.

## Definition of terms and operational definitions

Primary caregivers are family members or relatives who provide unpaid care and support to a person with mental illness [34].

Perceived stress refers to subjective feelings of stress experienced by primary caregivers of individuals with mental illness. The perceived stress was measured by the verbal responses to the Perceived Stress Scale (PSS-10); each of the ten items has a value of 0–4. The total scores range from 0 to 40 points. Items 4, 5, 7, and 8 were scored in a reverse manner. Scores categorized as 0–13 for low stress, 14–26 for moderate stress, and 27–40 for high stress [35].

## Data analysis

The data were entered into Epi-Data version 3.1 and exported to SPSS version 26 for analysis. Descriptive statistics were presented in the form of frequencies, tables, texts, and summary measures, and a multicollinearity test was

PLOS Mental Health

checked using the variance inflation factor (VIF) value, where a VIF less than ten was considered acceptable. Additionally, outlier and data normality tests were performed. Bivariate and multivariable ordinal logistic regression analysis was employed to identify the association between each independent variable and the outcome variable. All variables with a p-value < 0.25 in the bivariate ordinal logistic analysis were included in the final model of the multivariable ordinal analysis to control for all possible confounders, and ordinal logistic regression was used, which involved checking the model fitting information, goodness-of-fit, parameter estimates, pseudo R², and parallel line test to identify if the proportional odds model assumptions were violated and used the partial proportional odds model. These analyses were used to identify the predictors of perceived stress at a p-value <0.05. A 95% confidence interval or odds ratio was used to identify the significantly associated factors.

## Results

### Sociodemographics characteristics

Out of 422 primary caregivers, 409 participated in the study, resulting in a response rate of 96.9%. Most caregivers were male (64.1%), married (70.7%), and resided in rural areas (61.9%). The mean age of participants was 41.62 (SD±11.772) years. Among participants, 289 (70.7%) were married, and 285 (69.7%) were Muslim followers. 157 (38.4%) participants had no formal education, while others have education levels above elementary school. Additionally, 171 (41.8%) participants were farmers (Table 1).

### Clinical and psychosocial factors

The majority of patients were diagnosed with schizophrenia (54.5%), and 27.1% exhibited aggressive behavior. Caregivers with poor social support (69.2%) and high caregiving burden (36.2%) reported higher stress levels (Table 2).

### Perceived stress levels

The overall prevalence of perceived stress levels among primary caregivers of patients with mental illness showed significant variation. According to the findings, 13.20% (54) of caregivers indicated high levels of perceived stress (95% CI: 10.2-16.7), while 62.10% (254) reported moderate levels (95% CI: 57.3-66.7). Additionally, 24.69% (101) of primary caregivers reported experiencing low levels of stress (95% CI: 20.7-29.0). (Fig 1).

### Factors identified in the proportional odds model with the perceived stress of primary caregivers of patients with mental illness

The results from the multivariable ordinal logistic regression indicated that primary caregivers who were not educated, those dealing with patient aggressive behavior, high caregiving burden, low coping strategies, and patients with comorbid illnesses were significant factors associated with perceived stress (Table 3).

### Factors identified in the partial proportional odds model with the perceived stress of primary caregivers of patients with mental illness

Primary caregivers with poor social support exhibited significant differences in perceived stress levels, with high versus (low and moderate) perceived stress showing an odds ratio of OR = 5.18 (95% CI: 1.23, 12.74), and low versus (moderate and high) perceived stress showing OR = 0.19 (95% CI: 0.04, 0.80). The odds ratio of 0.19 indicates that primary caregivers with poor social support are less likely to report low perceived stress compared to those who report moderate or high perceived stress.

For primary caregivers with low resilience, the levels of perceived stress significantly differ between high and low and moderate perceived stress (OR = 3.09; 95% CI: 1.55, 6.15). However, there is a significant difference between low versus

**Table 1.** Socio-demographic and Economic Characteristics of Primary Caregivers of Patients with Mental Illness at Jimma Medical Center, Jimma, Southwest Ethiopia, 2024 (N = 409).

| Variable | Categories | Frequency (n) | Percentage (%) |
|---|---|---|---|
| Sex | Male | 262 | 64.1 |
| | Female | 147 | 35.9 |
| Age (mean 41.62 and SD = 11.772) | 18-24 | 15 | 3.7 |
| | 25-34 | 117 | 28.6 |
| | 35-44 | 119 | 29.1 |
| | 45-54 | 83 | 20.3 |
| | ≥55 | 75 | 18.3 |
| Ethnicity | Oromo | 307 | 75.1 |
| | Amhara | 42 | 10.3 |
| | Kefa | 23 | 5.6 |
| | Yem | 8 | 2.0 |
| | Dauro | 14 | 3.4 |
| | Other* | 15 | 3.7 |
| Marital status | Single | 78 | 19.1 |
| | Married | 289 | 70.7 |
| | Widowed | 27 | 6.6 |
| | Divorced/separated | 15 | 3.6 |
| Residence | Urban | 156 | 38.1 |
| | Rural | 253 | 61.9 |
| Religion | Muslim | 285 | 69.7 |
| | Orthodox | 74 | 18.1 |
| | Protestant | 43 | 10.5 |
| | Other** | 7 | 1.7 |
| Education level | Not educated | 157 | 38.4 |
| | Elementary (1–8) | 77 | 18.8 |
| | High school (9–12) | 90 | 22.0 |
| | College and above | 85 | 20.8 |
| Occupation | Farmer | 171 | 41.8 |
| | Merchant | 41 | 10.0 |
| | Government employment | 54 | 13.2 |
| | House wife | 54 | 13.2 |
| | Other*** | 89 | 21.8 |
| Average monthly Income **** | ≤3676ETB | 309 | 75.6 |
| | ≥3677ETB | 100 | 24.4 |

Notes: *Other ethnicity, Gurage, Siltee, Tigire.

Catholics and Jehovah's Witnesses.

***Retired, student.

**** Based on the World Bank, the global poverty lines were updated in September 2022.

moderate and high perceived stress (OR = 0.25; 95% CI: 0.13, 0.50). The odds ratio of 0.19 indicates that primary caregivers with low resilience are less likely to report low perceived stress by 74.1% compared to those who report moderate or high perceived stress.

**Table 2. Clinical Characteristics of Patients Attending the Psychiatric Outpatient Department at Jimma Medical Center, Jimma, Southwest Ethiopia, 2024 (N = 409).**

| Variables | Categories | Frequency (n) | Percent (%) |
|---|---|---|---|
| Psychiatric diagnosis | Schizophrenia | 223 | 54.5 |
| | Bipolar disorder | 97 | 23.7 |
| | MDD | 65 | 15.9 |
| | Other* | 24 | 5.9 |
| Comorbid medical illnesses | Yes | 41 | 10 |
| | No | 368 | 90 |
| Aggressive behavior | Yes | 111 | 27.1 |
| | No | 298 | 72.9 |
| Duration of illness in year | Mean 6.93 and SD (±5.683 years) | | |

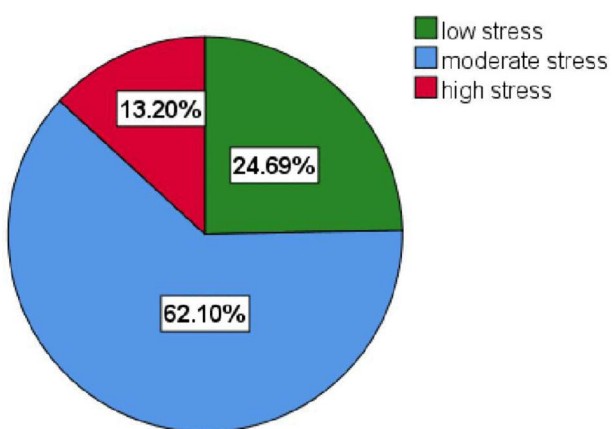

**Fig 1. Prevalence of perceived stress level among primary caregivers of patients with mental illness attending the psychiatric outpatient department at Jimma Medical Center, Southwest, Ethiopia, 2024 (N = 409).**

The odds ratio for low versus moderate and high stress is OR = 0.82 (95% CI: 0.76, 0.89), indicating that the likelihood of reporting low stress, as opposed to moderate or high stress, decreases by 17.2% for each unit increase in time spent per day with a patient. Conversely, the odds ratio for high stress versus low and moderate stress is OR = 1.22 (95% CI: 1.119, 1.32), suggesting that the likelihood of experiencing low or moderate stress increases by 22.6% for every unit increase in time spent per day with patients (Table 4).

## Discussion

This study revealed that 24.69%, 62.10%, and 13.20% of primary caregivers reported low, moderate, and high perceived stress, respectively. This result indicates that most primary caregivers reported a moderate level of perceived stress while caring for patients with mental illness. Caregivers who were not educated, those caring for patients with aggressive behavior and comorbid illnesses, and those experiencing high caregiver burden, low coping strategies, poor social support, low resilience, and time-spent caregiving were significantly associated with perceived stress.

**Table 3. Factors identified by a multivariable proportional odds model with perceived stress level among primary caregivers of patients with mental illness at Jimma Medical Center, Southwest Ethiopia, 2024.**

| Variable | Categories | Percent (%) | ^β | Sig. | OR | 95% CI(OR) | |
|---|---|---|---|---|---|---|---|
| | | | | | | LB | UB |
| Education level | Not educated | 38.4 | 0.805 | **0.021*** | 2.236 | 1.132 | 4.419 |
| | Elementary (1–8) | 18.8 | -0.304 | 0.407 | 0.738 | 0.359 | 1.516 |
| | High school (9–12) | 22.0 | 0.285 | 0.438 | 1.329 | 0.649 | 2.724 |
| | College & > **Ref.** | 20.8 | 0ª | . | 1 | . | . |
| Aggressive behavior | Yes = 1 | 27.1 | 0.840 | **0.004*** | 2.315 | 1.310 | 4.092 |
| | No = 2 **Ref.** | 72.9 | 0a | . | 1 | . | . |
| Caregiving burden | High = 0 | 36.2 | 1.389 | **<0.001** | 4.011 | 2.083 | 12.938 |
| | No&mild = 1 **Ref.** | 63.8 | 0ª | . | 1 | . | . |
| Coping strategies | Low | 4.4 | 1.284 | **0.003*** | 3.611 | 1.223 | 9.366 |
| | High = **Ref.** | 95.6 | 0ª | . | 1 | . | . |
| Comorbid illness | Yes | 10 | 1.123 | **0.000*** | 3.074 | 2.481 | 10.077 |
| | No = **Ref.** | 90 | 0ª | . | 1 | . | . |

* Significantly associated at p-value < 0.05; β is the coefficient.

**Table 4. Factors identified multivariable partial proportional odds model with perceived stress level among primary caregivers of patients with mental illness of Jimma Medical Center, Southwest Ethiopia, 2024.**

| Variable | characteristics | Percent (%) | COR1 (95% CI) | High Vs. (low and moderate stress) | | Low vs. moderate stress and high stress | |
|---|---|---|---|---|---|---|---|
| | | | | AOR1 (95% CI) | P-Value | AOR2 (95% CI) | p-value |
| Social support | Poor | 69.2 | 5.420 (1.809, 16.238) | **5.189** (1.238, 12.740) | **0.024*** | 0.193 (0.046, 0.808) | **0.024*** |
| | Moderate | 27.6 | 2.284 (0.745, 7.001) | 4.140 (0.967, 17.723) | 0.055 | 0.242 (0.056, 1.034) | 0.055 |
| | Strong **Ref.** | 3.2 | 1 | 1 | . | 1 | |
| Resilience | Low | 33.7 | 5.619 (2,955,106,820) | **3.097** (1.559, 6.150) | 0.000* | 0.323 (0.163, 0.641) | **0.001*** |
| | Normal | 52.3 | 1.846 (1.047, 3.254) | 1.496 (0.821, 2.727) | 0.189 | 0.873 (0.471, 1.618) | 0.666 |
| | High **Ref.** | 13.9 | 1 | 1 | . | 1 | |
| Time spent with patients. | | | 1.264 (1.173, 1.362) | **1.226** (1.119, 1.326) | 0.**000*** | 0.818 (0.754, 0.888) | **0.000*** |
| | (Scale) | | | | | | |

* Significantly associated at p-value < 0.05; β is the coefficient.

In the current study, a higher proportion of study participants reported moderate perceived stress, aligning with studies in the Indian state of Karnataka [36] and Indonesia [37] . However, these rates are lower than those found in another study in Indonesia [38], India [39–41], Nepal [42], and Bangladesh [43].

Additionally, the findings regarding high perceived stress are consistent with studies conducted in India [41] and Nepal [42] but are lower than those in studies conducted in India [40] and Indonesia [37,38] . However, the findings of the current study indicate higher rates than those reported in the Indian state of Karnataka, which has severe stress [44], and in Bangladesh [43].

The overall percentages suggest that moderate stress is a common experience among primary caregivers in different countries. Caregiving roles often lead to increased stress levels, although some stress levels vary slightly across countries. This discrepancy might be related to the fact that the classification of stress levels differs, while this study perceived stress categorized into three levels (low, moderate, and high perceived stress), which was a fair level of classification.

The study conducted in India used a self-rated questionnaire with purposive sampling. The data was collected by the Perceived Stress Scale-14 (PSS-14) tool, which was not modified, but this study used the PSS-10 and used face-to-face interviews. The study conducted in Nepal employed purposive sampling with a small sample size and utilized the Kingston Caregiver Stress Scale to collect data. In contrast, the present study used the PSS-10 with a larger sample size of primary caregivers of patients with mental illness. In a study conducted in the Indian state of Karnataka, a small sample was recruited using convenience sampling, and a 14-item stress scale was employed to classify levels of stress into normal, mild, moderate, severe, and extremely severe. Additionally, there are variations in socioeconomic status, healthcare accessibility, sample sizes, and cultural differences.

In this study uneducated primary caregivers were 2.236 times more likely to experience high stress than those with higher educational levels. This finding aligns with studies done in India [ 41,45] . This suggests that educational attainment can influence coping mechanisms and access to resources, which are crucial for managing stress. Caregivers lacking formal education may not have the knowledge or skills to effectively navigate the complexities of caregiving, leading to heightened feelings of being overwhelmed and stress.

This study reveals that primary caregivers for patients showing aggressive behavior were about 2.315 times more likely to experience a higher stress level than those with care for no aggressive behavior. This finding is consistent with studies in Tunisia [46], New York [47], and Indonesia [38]. This indicates that challenging behaviors, such as aggression, can exacerbate caregiver burden and lead to emotional distress [47]. The need for constant care and management of aggressive behaviors can create an environment of unpredictability, further contributing to caregivers' anxiety and stress.

The present study showed that primary caregivers with a high burden of caregiving were 4.011 times more likely to increase perceived stress than those with no or mild caregiver burden while caring for their family. This finding is supported by studies done in New York [47], United States [48], Singapore [49], India [50], Malaysia [51], and Australia [52]. The possible reason might be that most primary caregivers are not health professionals; hence, caregivers with high burdens may experience increased stress due to physical, emotional, and psychological demands; lack of support; and emotional exhaustion [48].

The study found primary caregivers of mentally ill patients with low coping strategies were 3.611 times more likely to experience elevated stress compared to those with effective coping strategies. This finding is in line with studies done in Egypt [53] and South India [45]. This suggests the importance of coping skills in managing the emotional, physical, and practical demands of caregiving. Ensuring caregivers have healthy coping techniques, such as stress management, problem-solving, and self-care skills, is crucial for their resilience and long-term sustainability.

Additionally, primary caregivers with poor social support are approximately 5.189 times more likely to experience high stress compared to those with low or moderate perceived stress. This finding aligns with studies done in New York [47], the United States [48], Singapore [49], Brazil [54], and the United States [55]. This suggests that social isolation can exacerbate feelings of loneliness and emotional distress among caregivers, and strong social support systems can provide emotional relief, practical assistance, and a sense of community, all of which are essential for mitigating caregiver stress.

In this study, primary caregivers with low resilience are about 3.097 times more likely to report high stress compared to those with low or moderate perceived stress. This finding is consistent with a study conducted in Tunisia [46] and Australia [52]. This might be possible due to the lack of effective coping mechanisms, poor emotional regulation, limited social support networks, a negative perception of challenges, and insufficient resource utilization for primary caregivers.

Finally, the time spent per day of primary caregivers on patients with mental illness is significantly associated with their stress levels. As the amount of care time increases, the likelihood of experiencing moderate or high stress also increases,

while lower caregiving time corresponds to lower stress levels. This finding is supported by a study conducted in Taiwan [56]. This suggests that longer caregiving hours may lead to diminished opportunities for self-care and personal time, which are crucial for maintaining mental health [57]. The emotional impact of caregiving, including feelings of sadness, frustration, guilt, and anxiety, can be exacerbated by prolonged caregiving without adequate breaks or support.

### Strengths

The strengths of this study include the presentation of results on perceived stress levels and the application of a robust partial proportional odds model of ordinal logistic regression to analyze ordered dependent outcome variables. This model enabled the disaggregation and grading of certain independent variables, despite the violation of the proportional odds assumption.

### Limitations

The study has limitations, including its reliance on face-to-face interviews, which may introduce recall bias, as well as the potential for social desirability bias, resulting in underreporting or overreporting of outcomes. Additionally, the use of a convenience sampling technique may introduce selection bias, as participants were recruited from caregivers attending the outpatient department during the study period. Therefore, the findings may not be fully representative of all primary caregivers of patients with mental illness in the wider community, which may limit the generalizability of the results. Furthermore, the study is unable to establish a cause-and-effect relationship between perceived stress and its predictors.

## Conclusion and recommendation

About three-fourths of primary caregivers of patients with mental illness perceived moderate to high levels of stress, while only one-fourth reported low perceived stress. Lack of caregiver's formal education, caring for patients with aggressive behavior, caring for patients with comorbid medical illnesses, high caregiver burden, low coping strategies, poor social support, low resilience, and the amount of time spent per day with patients are significantly associated with the perceived stress.

Stakeholders should create a supportive environment for primary caregivers of patients with mental illness to reduce their stress levels. Therefore, future researchers may conduct qualitative designs like grounded theory and case studies to explore caregiver burdens and longitudinal studies on the long-term mental health impact of caregiving.

## Supporting information

**S1 Data. Data.sav.**
(ZIP)

## Acknowledgments

The author expresses heartfelt gratitude to the study participants, data collectors, and advisors for their valuable contribution to the research work. We declare that this research article is our original work, has not been published in any journal, and that all sources of materials used for the article have been fully acknowledged.

## Author contributions

**Conceptualization:** Nuguse Daraje, Susan Anand, and Tilahun Legese.

**Formal analysis:** Nuguse Daraje, Susan Anand, Tilahun Legese.

**Funding acquisition:** Nuguse Daraje.

**Investigation:** Nuguse Daraje.

**Methodology:** Nuguse Daraje, Susan Anand, and Tilahun Legese.

**Project administration:** Nuguse Daraje.

**Resources:** Susan Anand, Tilahun Legese.

**Supervision:** Susan Anand, Tilahun Legese.

**Validation:** Susan Anand, Tilahun Legese.

**Visualization:** Nuguse Daraje.

**Writing – original draft:** Nuguse Daraje.

**Writing – review & editing:** Nuguse Daraje, Susan Anand, Tilahun Legese.

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
