## [Decision Letter · Decision Letter 0]

8 Dec 2025

PMEN-D-25-00350

Perceived Stress and Associated Factors among Primary Caregivers of Patients with Mental Illness attending Outpatient department at Jimma Medical Center, Southwest Ethiopia, 2024: A Cross-sectional Study

PLOS Mental Health

Dear Dr. Daraje,

Thank you for submitting your manuscript to PLOS Mental Health. After careful consideration, we feel that it has merit but does not fully meet PLOS Mental Health’s publication criteria as it currently stands. Therefore, we invite you to submit a revised version of the manuscript that addresses the points raised during the review process.

We look forward to receiving your revised manuscript.

Kind regards,

Ansar Abbas

Academic Editor

PLOS Mental Health

Journal Requirements:

1. Please provide a complete Data Availability Statement in the submission form, ensuring you include all necessary access information or a reason for why you are unable to make your data freely accessible. If your research concerns only data provided within your submission, please write "All data are in the manuscript and/or supporting information files" as your Data Availability Statement.

2. We have amended your Competing Interest statement to comply with journal style. We kindly ask that you double check the statement and let us know if anything is incorrect.

3. Please provide separate figure files in .tif or .eps format.

https://journals.plos.org/mentalhealth/s/figures

https://journals.plos.org/mentalhealth/s/figures#loc-file-requirements

4. We have noticed that you have uploaded Supporting Information files, but you have not included a list of legends. Please add a full list of legends for your Supporting Information files after the references list.

Reviewers' comments:

Reviewer's Responses to Questions

**Comments to the Author**

1. Does this manuscript meet PLOS Mental Health’s publication criteria? Is the manuscript technically sound, and do the data support the conclusions? The manuscript must describe methodologically and ethically rigorous research with conclusions that are appropriately drawn based on the data presented.

Reviewer #1: Partly

2. Has the statistical analysis been performed appropriately and rigorously?

Reviewer #1: Yes

3. Have the authors made all data underlying the findings in their manuscript fully available (please refer to the Data Availability Statement at the start of the manuscript PDF file)?

Reviewer #1: No

4. Is the manuscript presented in an intelligible fashion and written in standard English?

Reviewer #1: Yes

5. Review Comments to the Author

Reviewer #1: This manuscript investigates perceived stress and its associated factors among primary caregivers of patients with mental illness in Ethiopia. The topic is highly relevant to global mental health, and I commend the authors for focusing on an important and understudied population. However, several methodological, analytic, and reporting issues need to be addressed before the study can be considered for publication. My detailed comments are below.

1. Sampling and generalizability

The use of convenience sampling introduces potential selection bias. Please expand the Methods and Discussion to explain how this may affect the representativeness of the sample and the generalizability of the findings.

2. Scale reliability

The manuscript should report reliability statistics (e.g., Cronbach’s α) for the PSS-10 and other scales within this study sample. This will help readers assess measurement quality in your specific context.

3. Description of covariates

All covariates included in the bivariable and multivariable models should be clearly defined and justified in the Method section. Please describe how each variable was measured, coded, and selected for inclusion in the regression analyses.

4. Data availability

The Data Availability statement currently says “all data available,” but no dataset or repository link is provided. In line with PLOS policies, please either deposit the de-identified dataset in a public repository and include the link, or clearly explain any justified restrictions.

5. Paragraph structure and academic style

In several places, paragraphs are extremely short (one sentence) or fragmented. For academic writing, it would strengthen readability and flow to combine related sentences into fuller paragraphs (typically at least two to three sentences) that develop a clear idea.

6. Motivation and focus on caregivers

The Introduction would benefit from a clearer rationale for focusing specifically on primary caregivers of patients with mental illness rather than caregivers in general or patients themselves. Please expand on why this group is a critical target for research and intervention, and provide more contextual background for the Ethiopian setting.

7. Clarify this sentence in the Introduction

The sentence “However, there is lack of information on perceived stress levels and associated factors and can support inform interventions to reduce stress, improving patient outcomes and overall wellbeing of family.”

is difficult to follow and initially made it unclear whether the study focuses on caregivers or patients. I recommend rewriting this sentence to explicitly state that the study examines perceived stress among primary caregivers and how understanding these factors can inform caregiver-focused interventions.

8. Redundancy in Section 3.1

In Section 3.1 (Sociodemographic Characteristics), the proportion of male caregivers is mentioned twice. Please remove the duplicate description to avoid redundancy.

9. Measurement details for all scales

Recommend authors briefly describe for each: number of items, scoring range, cut-offs for “high” vs “low” (e.g., high burden, poor social support, low coping, low resilience), and provide references. This belongs in the Methods so readers can understand how those categorical variables in the regression were created.

Addressing these points will strengthen the clarity, rigor, and overall presentation of the manuscript.

6. PLOS authors have the option to publish the peer review history of their article (what does this mean?). If published, this will include your full peer review and any attached files.

**Do you want your identity to be public for this peer review?** For information about this choice, including consent withdrawal, please see our Privacy Policy.

Reviewer #1: No

Figure Resubmissions:

---

## [Decision Letter · Decision Letter 1]

4 Mar 2026

PMEN-D-25-00350R1

Perceived Stress and Associated Factors among Primary Caregivers of Patients with Mental Illness attending Outpatient department at Jimma Medical Center, Southwest Ethiopia, 2024: A Cross-sectional Study

PLOS Mental Health

Dear Dr. Daraje,

Thank you for submitting your manuscript to PLOS Mental Health. After careful consideration, we feel that it has merit but does not fully meet PLOS Mental Health’s publication criteria as it currently stands. Therefore, we invite you to submit a revised version of the manuscript that addresses the points raised during the review process.

We look forward to receiving your revised manuscript.

Kind regards,

Ansar Abbas

Academic Editor

PLOS Mental Health

Journal Requirements:

Additional Editor Comments (if provided):

Reviewers' comments:

Reviewer's Responses to Questions

**Comments to the Author**

1. If the authors have adequately addressed your comments raised in a previous round of review and you feel that this manuscript is now acceptable for publication, you may indicate that here to bypass the “Comments to the Author” section, enter your conflict of interest statement in the “Confidential to Editor” section, and submit your "Accept" recommendation.

Reviewer #1: (No Response)

2. Does this manuscript meet PLOS Mental Health’s publication criteria? Is the manuscript technically sound, and do the data support the conclusions? The manuscript must describe methodologically and ethically rigorous research with conclusions that are appropriately drawn based on the data presented.

Reviewer #1: Yes

3. Has the statistical analysis been performed appropriately and rigorously?

Reviewer #1: Yes

4. Have the authors made all data underlying the findings in their manuscript fully available (please refer to the Data Availability Statement at the start of the manuscript PDF file)?

Reviewer #1: Yes

5. Is the manuscript presented in an intelligible fashion and written in standard English?

Reviewer #1: Yes

6. Review Comments to the Author

Reviewer #1: 1. In your response to reviewers, you state that the Discussion/Limitations were updated to acknowledge potential selection bias from convenience sampling and its implications for representativeness/generalizability. However, in the revised manuscript I could not find an explicit statement addressing selection bias.

2. Please standardize country naming throughout the manuscript and correct the grammatical error “United State” to “United States.” For example, the Discussion currently uses mixed forms (e.g., “United State (33)” and “USA (40)”).

7. PLOS authors have the option to publish the peer review history of their article (what does this mean?). If published, this will include your full peer review and any attached files.

**Do you want your identity to be public for this peer review?** For information about this choice, including consent withdrawal, please see our Privacy Policy.

Reviewer #1: No

Figure Resubmissions:

---

## [Decision Letter · Decision Letter 2]

28 Apr 2026

Perceived Stress and Associated Factors among Primary Caregivers of Patients with Mental Illness Attending the Outpatient Department at Jimma Medical Center, Southwest Ethiopia: A Cross-sectional Study

PMEN-D-25-00350R2

Dear Daraje,

We are pleased to inform you that your manuscript 'Perceived Stress and Associated Factors among Primary Caregivers of Patients with Mental Illness Attending the Outpatient Department at Jimma Medical Center, Southwest Ethiopia: A Cross-sectional Study' has been provisionally accepted for publication in PLOS Mental Health.

Best regards,

Ansar Abbas

Academic Editor

PLOS Mental Health

Reviewer Comments (if any, and for reference):

Reviewer's Responses to Questions

**Comments to the Author**

1. If the authors have adequately addressed your comments raised in a previous round of review and you feel that this manuscript is now acceptable for publication, you may indicate that here to bypass the “Comments to the Author” section, enter your conflict of interest statement in the “Confidential to Editor” section, and submit your "Accept" recommendation.

Reviewer #1: All comments have been addressed

Reviewer #2: All comments have been addressed

2. Does this manuscript meet PLOS Mental Health’s publication criteria? Is the manuscript technically sound, and do the data support the conclusions? The manuscript must describe methodologically and ethically rigorous research with conclusions that are appropriately drawn based on the data presented.

Reviewer #1: Yes

Reviewer #2: Yes

3. Has the statistical analysis been performed appropriately and rigorously?

Reviewer #1: Yes

Reviewer #2: Yes

4. Have the authors made all data underlying the findings in their manuscript fully available (please refer to the Data Availability Statement at the start of the manuscript PDF file)?

Reviewer #1: Yes

Reviewer #2: Yes

5. Is the manuscript presented in an intelligible fashion and written in standard English?

Reviewer #1: Yes

Reviewer #2: Yes

6. Review Comments to the Author

Reviewer #1: (No Response)

Reviewer #2: all comments have been addressed.

7. PLOS authors have the option to publish the peer review history of their article (what does this mean?). If published, this will include your full peer review and any attached files.

**Do you want your identity to be public for this peer review?** For information about this choice, including consent withdrawal, please see our Privacy Policy.

Reviewer #1: No

Reviewer #2: **Yes:** Novita Intan Arovah
